Normal feeding movements expressed by dimensionality reduction of whole-body joint motions using principal component analysis

Nakatake Jun jyun_nakatake@med.miyazaki-u.ac.jp 1
Miyazaki Shigeaki 1
Arakawa Hideki 1
Chosa Etsuo 2
1 Rehabilitation Unit, University of Miyazaki Hospital , Miyazaki-shi , Miyazaki , Japan
2 Community Medical Center, University of Miyazaki Hospital , Miyazaki-shi , Miazaki , Japan
Young Jesse
Electronic publication date: 2025 Apr 25
Publication date: 2025
Volume: 13
Electronic Location ID: e19324
Received 2024 Dec 19; Accepted 2025 Mar 24
Copyright: ©2025 Nakatake et al.
Copyright year: 2025
Copyright holder: Nakatake et al.
License: This is an open access article distributed under the terms of the Creative Commons Attribution License, which permits unrestricted use, distribution, reproduction and adaptation in any medium and for any purpose provided that it is properly attributed. For attribution, the original author(s), title, publication source (PeerJ) and either DOI or URL of the article must be cited.
License URL: https://creativecommons.org/licenses/by/4.0/

Keywords: Principal component analysis, Occupational therapy, Rehabilitation, Kinematics, Biomechanics, Feeding, Eating, Activities of daily living, Movements, Postures

Funding: Clinical Research Support Grant from University of Miyazaki Hospital This study was supported by a Clinical Research Support Grant from University of Miyazaki Hospital. The funders had no role in study design, data collection and analysis, decision to publish, or preparation of the manuscript.

==============================
Background

Understanding elementary feeding movements and postures is essential for improving assessment and intervention strategies in occupational therapy, particularly for individuals with eating difficulties, and for educating caregivers and students. However, the current assessment tools lack precision in evaluating complex feeding movements and often rely on subjective judgments rather than objective measures. We aimed to determine elementary movements and postures corresponding to different feeding phases using principal component analysis (PCA).

Methods

This cross-sectional observational study was conducted at a Local National University Hospital and included 45 healthy, right-handed adult volunteers (23 men and 22 women) aged 20–39 years (mean age, 27.3 years), with no neurological or musculoskeletal impairments. Movements during yogurt feeding using a spoon were captured with a three-dimensional inertial sensor motion capture system. Principal components (PCs) and their scores were derived from PCA of whole-body joint motion data across four feeding phases. PC scores were compared between phases using Friedman’s and post-hoc tests.

Results

The primary PC, representing whole-body movement, accounted for 50.0% of the variance; the second PC, associated with hand direction changes, accounted for 13.7%. The cumulative variance of the first five PCs was 83.1%, including individual body-part movements and fixations or combinations of these. Significant differences existed between feeding phases, particularly in the reaching and transport phases, which showed greater whole-body movement than that during the spooning and mouth phases. Hand direction changes were more prominent during the spooning phase than during the mouth phase.

Conclusions

PCA helped determine key elementary movements and their corresponding feeding phases, which can be used to assess patients with feeding difficulties and guide occupational therapy interventions. Portions of this text were previously published as part of a preprint (https://doi.org/10.1101/2024.09.14.24313686).

Introduction

Eating difficulties caused by injuries or illnesses result in physiological, psychological, and social challenges (Cipriano-Crespo et al., 2020; Klinke et al., 2013). Impaired self-feeding skills are further associated with malnutrition (Ciliz et al., 2023). Occupational therapy addresses these issues by focusing on swallowing, posture, movement, equipment, care methods, and habits, as well as providing psychosocial interventions for patients (Boop, Smith & Kannenberg, 2017; Philipps et al., 2012). Several interventions have been developed to improve posture in children (Bhattacharjya et al., 2021; Mlinda, Leyna & Massawe, 2018) and provide intensive training for specific movements (Jo, Noh & Kam, 2020; Treger et al., 2012). Additionally, early rehabilitation using feeding devices has been introduced in intensive care units (Koester et al., 2018).

Eating evaluations are often conducted using assessment tools, such as the functional independence measure (Uniform Data System for Medical Rehabilitation, 1990) and the modified Barthel index (Shah, Vanclay & Cooper, 1989), which include eating as one of several daily tasks. Patients are assessed on an ordinal scale based on judgments regarding voluntary movements, caregiver support, or types of equipment used. An eating-specific screening tool, the minimal eating observation form II, has also been developed (Westergren et al., 2009). This tool consists of three items observed on a nominal or ordinal scale, focusing on the sitting position. Practitioners use these assessments to evaluate the eating conditions of patients, set treatment goals, and guide feeding movements and postures. However, targeted movements and postures may be assessed subjectively owing to the experience of practitioners, and the rationale for these assessments remains insufficient.

While studies have reported summarized measurements and waveforms regarding joint angles necessary for normal feeding activities (Nagao, 2004; Doğan et al., 2019; Van Andel et al., 2008), these movement and posture patterns correspond to different time phases (Nakatake et al., 2021). The results suggest substantial changes in whole-body joint angles during the phases of reaching for the dish and transporting food to the mouth. Additionally, motion direction varies during the phases of spooning food and taking it into the mouth. Understanding these joint motions, which involves changes in each joint angle over time, can enhance assessments and interventions aimed at improving the movements or postures of patients (Kontaxis et al., 2009). However, relying solely on individual joint motion attached to corresponding time phases to understand feeding movements is insufficient. Practitioners often recognize the movements of patients based on approximate body part movements rather than individual joint motions, which may involve a combination of joint motions, namely, coordination. These movements during feeding—such as reaching for food, manipulating the direction of the palm, approaching the food with the mouth, or stabilizing the trunk to support the upper limbs (Boop, Smith & Kannenberg, 2017; Philipps et al., 2012)—may involve coordinated, complex joint motions that play significant roles in daily living. However, their specific functions are derived from empirical knowledge and are not yet fully understood in scientific terms.

To address this issue, we employed principal component analysis (PCA) with quantitative kinematic measurements. This method highlights specific fictitious features of data through dimensional reduction, summarizing variables and removing redundancies in the dataset, which is advantageous for analyzing rich biomechanical variables (Daffertshofer et al., 2004). Kinematic data encompass multiple dimensions, making it challenging to interpret targeting movements, especially during the exploratory stage of research in this field. Therefore, PCA has revealed elementary movement (EM) patterns from gross movement presented with kinematic data during tasks, such as reaching (Kaminski, 2007) and trunk bending (Tricon et al., 2007). Furthermore, complex movements, such as bilateral upper-limb movements (Burns et al., 2017) and postures in sign languages (Bigand et al., 2021) have been categorized into patterns. EM is interpreted as a combination of joint motions for movements that interact with the environment or within the body.

We hypothesized that EMs could be defined from combinations of joint motions during feeding movements and that the appearance of EMs would differ between feeding phases. Therefore, we assessed the secondary analysis of a previous dataset (Nakatake et al., 2021) and aimed to determine the EMs involved in the feeding phases of whole-body joint motion in healthy individuals using PCA. The identified normal feeding movements and postures could provide clinical observational assessments or intervention cues for patients with eating difficulties.

Materials & Methods

Participants

The study adhered to the STROBE guidelines (Von Elm et al., 2007). The sample for this study was included in a previous study (Nakatake et al., 2021), and the study protocol was approved by the Research Ethics Committee of the Faculty of Medicine at the University of Miyazaki (Miyazaki-shi, Japan) (approval number: O-1501). Because we could not contact previous study participants, they were informed about the study through the institution’s website and provided with the option to opt out of participation at any time. Consequently, informed consent was indirectly obtained from participants who did not decline to participate in this study, although no written consent forms were collected. Furthermore, the authors did not have access to information that could identify individual participants of the previous study after data collection. In total, 50 participants were recruited from our institutional staff from April 2013 to October 2017, meeting the following criteria: aged 20–39 years, right-handed, and without neurological or musculoskeletal impairments. Individuals who were left-handed for regular spoon use were excluded.

Measurement procedures

The measurement procedures performed in the occupational therapy room at the institution and instrument details have been previously described (Nakatake et al., 2021). The feeding movements of participants were recorded using a three-dimensional motion capture system (Xsens MVN system; Xsens Technologies B.V., Enschede, Netherlands (Roetenberg, Luinge & Slycke, 2013)). This system provides kinematic output of a biomechanical whole-body model composed of 17 inertial sensors (with six degrees of freedom for positions and orientations in three-dimensional space) attached to the participant’s head, sternum, scapulas, pelvis, upper arms, forearms, hands, upper legs, lower legs, and feet (Fig. 1), consisting of 23 body segments, including the head, neck, pelvis, four vertebrae, scapulae, upper arms, forearms, hands, upper legs, lower legs, feet, and toes. The neutral (zero) position of the joint angles was defined as the joint angle when standing upright with feet parallel, one foot width apart, upper limbs alongside the body, palms facing forward, and the head oriented forward.

Figure 1 Inertial sensor position attached to participants.

(A) Front view. (B) Rear view. White squares indicate sensor position. The sensor is attached to the participant using a Lycra suit, a head band and gloves.

Participants sat on a stool of height 40 cm without a backrest, with a table adjusted at their elbow height that were positioned in front of their trunk at a distance of 10 cm (Fig. 2). They were instructed to use their right hand to reach for yogurt in a bowl placed on the table, scoop it with a stainless spoon (17.5 cm in length, 41 g in weight), transport it, and bring it to their mouth at a comfortable pace, repeating the sequence thrice. The aim was to capture voluntary movements; therefore, movements from after the spoon left the mouth to before the second instance, and from after the second instance to before the third instance, were analyzed.

Figure 2 Experiment setting for the feeding task.

The picture is the starting position of a feeding task. Participant has a spoon with right hand. The camera and laptop personal computer are set on the left side of the picture.

Sample selection

Given that participants performed their own feeding movements, five individuals displaying the following movements were excluded from the standardization of normal feeding movements: unnecessary upper limb elevation, shaking yogurt off the spoon while transporting it to the mouth, looking away, shaking the head vertically while reaching for the bowl, repeated spooning, or separating yogurt with the spoon. The final sample included 45 participants (23 men and 22 women) with a mean age of 27.3 years (standard deviation (SD) = 5.1) and a mean height of 164.8 cm (SD = 8.6).

Data analysis

Joint angles during a successive feeding cycle, consisting of reaching the hand to the bowl (reaching phase), spooning yogurt (spooning phase), transporting yogurt to the mouth (transport phase), and bringing yogurt to the mouth (mouth phase), were identified by confirming pictures in recorded movies synchronized to the system and extracted from the biomechanical model. Data were collected for the right shoulder (flexion/extension, abduction/adduction, and internal/external rotation), elbow (flexion/extension), forearm (pronation/supination), wrist (palmar/dorsal flexion and radial/ulnar deviation), C7-T1 (flexion/extension and right/left lateral flexion), and hip (flexion/extension) at a frequency of 120 Hz. A typical case is displayed in Fig. 3. The change in joint angles in each feeding phase was calculated using maximum and minimum values. Performance times during the phases were also recorded. These parameters or combinations of these parameters could explain human movements and postures (Kontaxis et al., 2009).

Figure 3 Typical waveforms of whole-body joint angle change across all feeding phases.

(A) Changes in the shoulder and elbow joint angles. (B) Changes in the forearm and wrist joint angles. (C) Changes in the neck (C7-T1) and hip joint angles. In all figures, the vertical axis indicates the joint angle, the horizontal axis indicates the performance time, and the vertical dotted line indicates the initial and final durations of each feeding phase.

This study performed PCA, which summarizes entire variables to theorical fictitious components called principal components (PCs), removing PCs involving unnecessary data variance. PCs comprise the sum of substantial measurements loaded by coefficients. Accordingly, each data point is associated with PC scores. To test our hypothesis, we analyzed changes in joint angles and performance times (11 variables) for one sample containing four feeding phases for each participant (data points of 45 × 4 = 180) using PCA. Subsequently, a PC was selected to ensure that the cumulative variance ratio (contribution ratio to all PC) was ≥80%, which was intended for the inclusion of as many as possible PCs because feeding involves various movements and postures in some phases. The Kaiser rule, which involves rejecting PCs with eigenvalues of <1, was not applied to avoid rejecting PCs that represent functional movements or postures related to normal feeding but have insufficient variance. Additionally, a scree plot of eigenvalues was visualized for supplementary PC selection. PCs with flatter slopes were not selected for unnecessary variance of entire data, while other PCs before them were chosen. PCs indicating EMs were interpreted from parameters having the loading of ≥|0.4|. Loaded parameters related to joint angle changes and performance times were considered with a plus or minus sign indicating event promotion or demotion, respectively. Furthermore, recent kinematic research on whole-body movements during reaching and transporting feeding phases (Nakatake et al., 2021) and practitioner’s empirical knowledge such as that of the ability to see foods and utensils, for proximal postural control for upper-limb and oral functions, for mouth orientation and manipulation toward food, for hand manipulation, and for upper-limb mediation the use of spoons for foods and the body itself in some feeding phases (Boop, Smith & Kannenberg, 2017; Philipps et al., 2012) were referred to validate our selected PCs comprising kinematic parameters loaded as having functional roles in daily feeding. Additionally, PC scores were calculated and compared between feeding phases using the Friedman test, with the significance level set at p < .05. Post-hoc analysis was performed using the Wilcoxon signed-rank test with Bonferroni correction (p < .008). The effect size r was interpreted as medium = |0.3| and large = |0.5| (Coolican & Coolican, 2013).

Results

PCA

Actual measurement data for joint angle changes are available in a previous paper (Nakatake et al., 2021) and those for performance times in Table S1. PCA of these kinematic data showed that the first PC accounted for 50.0% of the variance (eigenvalue, 5.50), followed by the second PC accounting for 13.7% (1.51). The variances for PCs 3–5 were 7.8% (0.86), 6.5% (0.72), and 5.0% (0.55), respectively. The cumulative variance across the first five PCs totaled 83.1%. Figure 4 shows the scree plot of eigenvalues for all PCs, indicating that PC 6 and beyond form a plateau. Thus, only the first five PCs were selected for further analysis.

Figure 4 Scree plot of the eigenvalue of PCs.

PC, principal component.

The loadings of ≥ |0.4| for each PC (Table 1) allowed for the following interpretations: PC 1 represented whole-body movement over time, assuming the hand leaving from the mouth during the reaching phase and the hand nearing the mouth during the transport phase; PC 2 indicated changes in hand direction while maintaining head stability, which was estimated because the hand controls a spoon for directing, taking up, and holding foods and for maintaining the view of foods and mastication; PC 3 involved elbow joint motion with stable shoulder joint angles, enabling upper-limb reaching and transporting while keeping the proximal body part stable; PC 4 captured lateral neck motion with fixed elbow angles, which allows for the imaging of the mouth approaching, taking up foods, and leaving from a spoon; and PC 5 reflected wrist palmar/dorsal flexion, which may facilitate to uptake and insertion of foods into the mouth.

Table 1 Parameter loadings of PCs.

	PC 1	PC 2	PC 3	PC 4	PC 5	
Shoulder Flex/Ext	0.82	−0.14	−0.44	−0.24	−0.01	
Shoulder Abd/Add	0.84	−0.05	0.05	0.11	0.07	
Shoulder In/Ex rotation	0.74	0.08	−0.57	0.06	−0.21	
Elbow Flex/Ext	0.64	−0.36	0.41	−0.44	0.00	
Forearm Pro/Sup	0.81	0.24	0.09	−0.30	−0.24	
Wrist Pal/Dors flexion	0.66	0.41	−0.06	0.06	0.59	
Wrist Rad/Uln deviation	0.44	0.74	0.22	−0.08	−0.14	
C7-T1 Flex/Ext	0.63	−0.56	−0.05	0.10	0.13	
C7-T1 R/L lateral flexion	0.69	−0.20	0.24	0.54	−0.24	
Hip Flex/Ext	0.72	−0.30	0.17	0.00	0.09	
Performance time	0.70	0.37	0.13	0.22	−0.01	
Notes.

Bold text indicates loadings of ≥|0.4|. PC, principal component.

Movements characterizing feeding phases

Friedman’s test revealed significant differences in PC scores across feeding phases for PCs 1–5 (PC 1, χ2 = 122, p < .0001; PC 2, χ2 = 109, p < .0001; PC 3, χ2 = 32, p < .0001; PC 4, χ2 = 66, p < .0001; PC 5, χ2 = 18, p = .0004).

The PC scores for each phase across the first five PCs are summarized in Fig. 5 and Tables S2 and S3. PC 1 scores were highest in the reaching phase, followed by the transport, spooning, and mouth phases, with significant differences and large to medium effect sizes between phases (reaching vs. spooning, z =  − 5.8, p < .0001, r =  − 0.62; reaching vs. transport, z =  − 5.8, p < .0001, r =  − 0.61; reaching vs. mouth, z =  − 5.8, p < .0001, r =  − 0.62; spooning vs. transport, z =  − 5.8, p < .0001, r =  − 0.62; spooning vs. mouth, z =  − 3.5, p = .0005, r =  − 0.36; transport vs. mouth, z =  − 5.8, p < .0001, r =  − 0.62).

Figure 5 Comparison of PC scores between feeding phases.

Vertical and horizontal axes indicate the PC score and feeding phases, respectively. The upper, middle, and lower lines of the boxplot indicate the upper, median, and lower quartiles, respectively. The upper and lower bars indicate the maximum and minimum values, respectively. Dots indicate outliers. Box plots connected by the above lines show significant differences between feeding phases, as determined using post-hoc analysis of the Wilcoxon signed–rank test with Bonferroni correction (p < .008). PC, principal component.

For PC 2, scores between all phases were significant with large effect to medium sizes, in the order of spooning, reaching, mouth, and transport phases (reaching vs. spooning, z =  − 3.1, p = .0018, r =  − 0.33; reaching vs. transport, z =  − 5.8, p < .0001, r =  − 0.62; reaching vs. mouth, z =  − 5.5, p < .0001, r =  − 0.58; spooning vs. transport, z =  − 5.8, p < .0001, r =  − 0.62; spooning vs. mouth, z =  − 5.8, p < .0001, r =  − 0.62; transport vs. mouth, z =  − 4.9, p < .0001, r =  − 0.52).

PC 3 scores were higher in the reaching, spooning, and transport phases than in the mouth phase, with large to small effects (reaching vs. mouth, z =  − 2.7, p = .0066, r =  − 0.29; spooning vs. mouth, z =  − 5.5, p < .0001, r =  − 0.58; transport vs. mouth, z =  − 3.1, p = .0019, r =  − 0.33).

For PC 4, the mouth phase had the highest scores (mouth vs. reaching, z =  − 2.8, p = .0045, r =  − 0.30; mouth vs. spooning, z =  − 5.6, p < .0001, r =  − 0.59; mouth vs. transport, z =  − 5.6, p < .0001, r =  − 0.59), whereas the transport phase had the lowest scores (transport vs. reaching, z =  − 5.2, p < .0001, r =  − 0.55; transport vs. spooning, z =  − 4.0, p = .0001, r =  − 0.43), with significant differences and large or medium effect sizes.

Finally, PC 5 scores were significantly higher in the spooning and transport phases than in the reaching phase, with medium effect sizes (spooning vs. reaching, z =  − 3.3, p = .0009, r =  − 0.35; transport vs. reaching, z =  − 3.3, p = .0009, r =  − 0.35).

Discussion

In this study, we aimed to identify elementary feeding movements and postures based on joint kinematics using PCA and to compare these movements across different feeding phases. The analysis revealed that the five PCs accounted for over 80% of the variance across all phases, supporting the hypothesis that EMs are defined by combinations of joint motions during feeding and that their occurrence varies across feeding phases. These findings suggest that understanding EMs can enhance the ability of occupational therapists to assess and improve feeding movements and postures through targeted interventions, such as positioning, specific movement training, and the use of adaptive devices.

The primary EM, involving whole-body movement for mouth and hand coordination, was most prominent in the reaching phase, followed by the transport phase. The second EM, which involved changes in hand direction by coordinating wrist joint motions with the fixed neck flexion angle, was prominent in the spooning phase followed by the reaching phase. The third EM, characterized by elbow motion with fixed shoulder angles, was frequently observed in the spooning, transport, and reaching phases. Lateral neck motion with fixed elbow angles was mostly observed in the mouth phase, but not in the transport phase. The spooning and transport phases involved more wrist flexion/extension movements (Fig. 6).

Figure 6 Image of PCs embedded in feeding phases.

(A) Reaching phase. (B) Spooning phase. (C) Transport phase. (D) Mouth phase. The line art shows human movements and postures while sitting during each feeding phase. The arrow directions indicate the possibility of motion corresponding to each body part. Red circles with oblique lines indicate that the motion of the overlapping arrows does not occur. Light blue indicates whole-body movements that require time during the reaching and transport phases. Green indicates changing the hand direction while maintaining the neck flexion angle during the reaching and spooning phases. Orange indicates the elbow motion that maintains shoulder flexion and rotation angles during the reaching, spooning, and transporting phases. Pink indicates lateral neck motion while maintaining the elbow angle in the mouth phase. Gray indicates wrist palm/dorsal flexion during spooning and transport phases. PC, principal component.

The whole-body movement observed in the reaching and transport phases aligns with that reported previously (Nakatake et al., 2021), confirming the coordination of upper and lower limb and neck joint motions. This theorical EM was revealed by the application of PCA to the biomechanical data. During the reaching phase, the shoulder flexes, abducts, and internally rotates, whereas the elbow extends, positioning the hand toward the bowl. The transport phase is characterized by the movement of the upper limb toward the mouth. The range of motion in hip flexion/extension during reaching facilitates the trunk’s return to a neutral position, which is necessary for hand-reaching motion within upper limb length (Kaminski, Bock & Gentile, 1995). Additionally, head and trunk movements during the transport phase bring the mouth closer to the food (Chinju et al., 2024; Inada et al., 2012; Van der Kamp & Steenbergen, 1999). The coordination of arm, neck, and trunk motions establishes the coupling of whole-body movements across the two feeding phases. The other PCs were defined as theoretical EMs for the first time. Notably, PC 3 indicated shoulder joint fixation in most feeding phases, highlighting the stabilization of proximal body parts. Shoulder stability is crucial for the functional performance of the upper extremities in individuals with disabilities (Olczak, Truszczyńska-Baszak & Mróz, 2022). Similarly, healthy feeding likely requires stabilizing the upper arm.

Our PCA revealed that various body segment movements dimensionally reduced the configuration of normal feeding: PC 2 and PC 3 in the reaching phase, PC 2, PC 3, and PC 5 in the spooning phase, PC 3 and PC 5 in the transport phase, and PC 4 in the mouth phase. These upper-body joint motions, which configure each PC in the corresponding feeding phases, have been confirmed in recent research (Nakatake et al., 2021). Integrating empirical knowledge (Boop, Smith & Kannenberg, 2017; Philipps et al., 2012), each EM can be interpreted as follows: PC 2, which involves positioning the hand by changing the two coupled wrist joint motions, represents the direction and manipulation toward foods on a table. These coincide with head stability for looking at objects and fine mastication. PC 3 reflects hand transport away from the trunk or closing the mouth via elbow joint motions, representing an upper limb reaching movement. These movements are well-established functions of the upper extremities (Kapandji, Owerko & Anderson, 2019). Additionally, the elbow joint motion of PC 3 may be adapted for spooning yogurt. These movements are based on shoulder joint stability. Counterintuitively, food intake into the mouth involved neck lateral flexion with elbow joint fixation (PC 4). Wrist flexion/extension (PC 5) enables manipulating and transporting foods, with these aspects warranting consideration.

The study results suggest that normal feeding involves various elements related to neck, trunk, and upper extremity movements and postures across different feeding phases. This objective knowledge clarifies our practical experience and supports more effective interventions for patients with eating difficulties, as outlined in the following implications. Practitioners may assess whether the feeding movements of patients are within normal ranges using EMs identified through PCA in this study. Treatment goals and programs focused on specific EMs can be developed to address feeding difficulties. To enhance feeding movements and postures, practitioners might consider implementing targeted positioning strategies, intensive movement training, or the use of adaptive devices, along with providing education for patients and their caregivers or occupational therapy students.

This study has some limitations, including its focus on healthy, right-handed individuals using a spoon to eat yogurt, because of the heterogeneity of dominant upper-limb movements between right- and left-handed individuals (Nelson, Berthier & Konidaris, 2018). These specific conditions may influence the observed EMs and warrant further research to better understand and validate these findings. This method may uncover new EMs, such as trunk stability, because patient instability impacts upper-limb functions (Olczak, Truszczyńska-Baszak & Mróz, 2022). Individuals exhibiting EMs outside the normal range, such as the minimum or maximum values of PC 1 during the reaching phase, should be noted in clinical assessments. In addition, some different PCs had a common parameter, which can be explained as this parameter existing in different phases; however, the common parameter of different PCs in the common feeding phase may indicate within or between subject differences and should be clarified in future studies. Furthermore, while this study revealed the relationship between kinematic PCs (EMs) and feeding phases, the role/function/intension of movements and postures shown using PCs is an assumption, which should be considered during its interpretation. In this study, we employed a portable inertial motion capture system, which offers flexibility in measurement settings. However, alternative methods, such as single-camera markerless capture (Scott et al., 2022) or visual kinematic observation (Bernhardt, Bate & Matyas, 1998), may offer additional benefits in clinical practice (Demers & Levin, 2017) and should be evaluated for their applicability.

Conclusions

The PCA of whole-body kinematic data identified several EMs associated with normal feeding, each corresponding to specific functional phases. Moreover, these EMs, including whole-body movement and distinct body part movements or postures, were assumed for movements such as closing and releasing the hand from the mouth, the hand using a spoon to direct and take up foods, and the mouth taking up foods. These findings provide a theoretical foundation for defining normal feeding movements and postures. Further research is warranted to validate the application of these findings to clinical practices related to addressing feeding difficulties.

Supplemental Information

Supplemental Information 1 Performance time analyzed using principal component analysis

Raw performance times and subject codes.

Supplemental Information 2 Principal component (PC) score for each feeding phase

Values are presented as median (interquartile range). PC, principal component.

Supplemental Information 3 Post-hoc comparison of principal component (PC) scores between feeding phases

The comparison was performed using Wilcoxon signed-rank test with Bonferroni correction (p < .008). PC, principal component.

Additional Information and Declarations

Competing Interests

Author Contributions

Human Ethics

Data Availability

The authors declare there are no competing interests.

Jun Nakatake conceived and designed the experiments, performed the experiments, analyzed the data, prepared figures and/or tables, authored or reviewed drafts of the article, conceptualization, and approved the final draft.

Shigeaki Miyazaki analyzed the data, authored or reviewed drafts of the article, validation, and approved the final draft.

Hideki Arakawa analyzed the data, authored or reviewed drafts of the article, funding acquisition, and validation, and approved the final draft.

Etsuo Chosa conceived and designed the experiments, authored or reviewed drafts of the article, conceptualization, project administration, supervision, and validation, and approved the final draft.

The following information was supplied relating to ethical approvals (i.e., approving body and any reference numbers):

The Research Ethics Committee of the Faculty of Medicine at the University of Miyazaki approved the study protocol.

The following information was supplied regarding data availability:

The raw measurements and subject codes for joint angle changes are available at PONE: https://doi.org/10.1371/journal.pone.0259184.s005.

The performance times and subject codes are available in the Supplemental Table.

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
