# Peer review of "Normal feeding movements expressed by dimensionality reduction of whole-body joint motions using principal component analysis"

_PeerJ, doi:10.7717/peerj.19324_

## Round 0.1 · original submission · Major Revisions

Two external peer reviewers have evaluated this manuscript. Both reviewers are receptive to the aim of the study, and think that the findings are generally important. However, both also identify several areas where the methods and results sections can be improved. Specifically, reviewers note issues with the implementation, presentation, and interpretation of the principal components analysis. I urge the authors to make careful consideration of all their points.

Reviewer 1 ·

Basic reporting

The manuscript titled “Normal feeding movements expressed by dimensionality reduction of whole-body joint motions using principal component analysis” is highly interesting and provides a unique perspective on the study of elementary movements. However, there are two major issues that the authors must address before the manuscript can be considered for publication.

1) Absence of Proper Determination of Principal Components
In the section describing the PCA methodology, the authors state that they selected the number of principal components solely based on the cumulative variance criterion (≥85%). However, they do not mention the use of standard statistical methods for determining the optimal number of PCs, such as the Scree Plot (Elbow Method) or the Eigenvalue Criterion (Kaiser’s Rule, Eigenvalues >1). The cumulative variance criterion alone is insufficient for justifying PC selection, as it risks including more PCs than necessary, increasing complexity without improving the interpretability of the results. The lack of these methods presents two major concerns:
- It is unclear whether PC5 and PC6 were truly necessary, or if the analysis should have been limited to fewer components (e.g., PC3 or PC4).
- There is no verification that the selected components have eigenvalues greater than 1, which would indicate that some of them might not be informative and should be excluded from the analysis.
To ensure methodological rigor, the authors should include a Scree Plot to visualize the explained variance and confirm the selection of PCs, as well as report the eigenvalues to justify the retention of each component.

2) Unclear Justification for PC Interpretation
The paper states that PC1 represents "whole-body movement over time", and similarly assigns specific biomechanical meanings to other PCs (e.g., PC2 captures changes in hand direction while maintaining head stability; PC3 captures elbow motion with stable shoulder angles). However, the methodology used to support these interpretations is unclear. If PC2, PC3, and other components capture distinct motion patterns, it raises further questions about why PC1 alone is considered to represent whole-body movement. The interpretation of PCs should be explicitly based on objective criteria, such as biomechanical studies, kinematic analyses, or the distribution of PC loadings across all joint angles.
Therefore, the authors should clarify the reasoning behind labeling PC1 as "whole-body movement" and provide similar justifications for all other PC interpretations. This will enhance the transparency and validity of the analysis.

Final Recommendation
To improve the clarity and rigor of the study, the authors should:
- Provide a Scree Plot to confirm the number of selected PCs.
- Report the eigenvalues of each PC to ensure their statistical significance.
- Clearly justify the interpretation of PC1 and other PCs based on biomechanical principles and PC loadings.

Experimental design

Please refer to the previous comments.

Validity of the findings

Please refer to the previous comments.

Additional comments

Please refer to the previous comments.

Reviewer 2 ·

Basic reporting

The article is well presented especially in the introduction of the problem of normal feeding, and the review of the relevant literature in the field. It is a very unique problem and although it does not seem to receive much attention in the literature as one would have expected for a topic of this nature, it can simply be due to oversight. It is an impactful area of research.
There are though some areas that I feel should have been presented much better than they appear in the manuscript. These are include the following:
1. The presentation and discussion of the analysis tools that the authors used is quite sketchy. I am aware that principal components analysis is a well-known and established tool, but I think it is better to discuss with some level of detail on how it is being applied in the problem. Its exact usage is quite hard to understand and therefore it become hard to appreciate its usefulness. It makes the application hard to be replicated or applied to related problems where such tool could be useful.
2. The results section seems to focus much on the analysis results and not necessarily the measured data as well. I was also over the view that the presentation could have been better if the authors presented the data in tabular format and then save textual content to explain their results rather than simply listing down values and analysis results.
3. The conclusion section would have benefitted from more discussion on how well the project achieved its goals. It is too brief at the moment.

Experimental design

The experimental design is fairly well done, although it would have benefitted from some pictures of the actual measurement set-up rather than simple sketches. It loses some level of credibility by simply using sketches. I think that if this was done on matters of confidence to the participants, the authors could have concealed their faces and all areas that could exposed them to easy identification.
The exact placement of the sensors and the direction and angles of measurement should be clearly illustrated in the experimental design. I feel that the authors have no sufficiently done this.

Validity of the findings

The findings are sound and seem valid in spite of the above highlighted shortcomings. The finds are properly discussed in the "Discussion". I would urge the readers to attend to the format of the results presentation so that they focus on explaining the technical details rather than simply documenting the analysis results.
As already mentioned, the findings should include a presentation of the actual the actual measured data.

Additional comments

None

---

## Round 0.2 · Minor Revisions

I thank the authors for their work responding to the previous round of reviews. Reviewer 1 has one remaining small suggestion for improving the clarity of the statistical analyses. Once the authors respond to this suggestion, I'd be happy to accept the manuscript.

Reviewer 1 ·

Basic reporting

The authors addressed the comments correctly. The explanation of the PCA analysis is much improved, allowing readers to understand your approach and potentially apply it to assessing other functional movements.

Experimental design

The research question is clearly defined and the methods are clearly explained.

Validity of the findings

The findings correctly point out the meaning of the manuscript.

Additional comments

A minor comment: since choosing between the Kaiser rule and the 80% variance criterion remains a dilemma in PCA, please clarify why the authors preferred using the 80% variance criterion (i.e., because you believed that PC3–5 contained additional valid information for the feeding movement) rather than following the eigenvalue threshold.

---

## Round 0.3 · accepted · Accept

I thank the authors for addressing all remaining questions, by clarifying statistical methods. I’m happy to recommend acceptance.